# Quantifying gender gaps in seismology authorship

Laura Ermert[1, *], Maria Koroni[1, *], and Naiara Korta Martiartu[2, *]

[1]Swiss Seismological Service, ETH Zürich, Zürich, Switzerland
[2]Institute for Applied Physics, University of Bern, Bern, Switzerland
[*]These authors contributed equally to this work and are ordered alphabetically.

**Correspondence:** Laura Ermert (laura.ermert@sed.ethz.ch), Maria Koroni (maria.koroni@sed.ethz.ch), and Naiara Korta Martiartu (naiara.korta@unibe.ch)

**Abstract.** According to 2018 demographic data of the American Geophysical Union Fall Meeting, seismology is among the Geoscience fields with the lowest representation of women. To understand whether this reflects seismology more generally, we investigate women's authorship of peer-reviewed publications, a key factor in career advancement. Building upon open-source tools for web-scraping, we create a database of bibliographic information for seismological articles published in 14 international journals from 2010 to 2020. We use the probabilities of author names being either male- or female-gendered to analyse the representation of women authors in terms of author position and subsequently per journal, year, and publication productivity. The results indicate that: 1) The overall probability of the first (last) author being female is 0.28 (0.19); 2) With the calculated rate of increase from 2010 to 2020, equal probabilities of female and male authorship would be reached towards the end of the century; 3) Compared to the overall probability of male authorship (0.76), single-authored papers in our database are disproportionately published by male authors (with probability 0.83); 4) Female representation decreases among highly productive authors; 5) Rather than being random, the composition of authorship appears to be influenced by gender: firstly, all-male author teams are more common than what would be expected if teams were composed randomly. Secondly, the probability that first or co-authors are female increases when the last author is female, but first female authors have a low probability of working with female co-authors.

## 1 Introduction

In seismology, as in many fields of research, peer-reviewed articles are one of the most important ways to disseminate new scientific findings. They are also increasingly used as a metric of performance and productivity of individual researchers and constitute a critical factor of career advancement, along with citation scores and the impact factor of journals where researchers publish (West et al., 2013). Gender inequality, especially in higher-level academic positions, is a persistent problem in the fields of science, technology, engineering, and mathematics (STEM) and in academia more generally. The attrition of women graduates has been described as a leaky pipeline, with women dropping out at higher rates than men at various career stages. Data collected by UNESCO show that while women graduate from any field of study with BSc and MSc at slightly higher rates than men globally (with 53 and 55 % women graduates, respectively), they are slightly less well represented at the PhD level (44 %) and significantly less well represented at the researcher level (29 %), with very pronounced regional

variations (Fernandez Polcuch et al., 2018). The European Commission reports that, in research in general, the representation of women drops from 52 % at the PhD level to 26 % at the highest career level, while for STEM, 38 % of PhDs and 19 % of senior faculty are women (European Commission, Directorate-General for Research and Innovation, 2021, p. 182).

The leaky pipeline effect has also been documented in the geosciences. Agnini et al. (2020) investigated the situation in Italy and reported a drop of women geoscientists from around 50 % at the PhD level to around 20 % at the full professor level (data from 2012 and 2014). Holmes et al. (2008) and Ranganathan et al. (2021) found a similar picture at US-American universities, where approximately 45 % of graduate students and below 15 % of full professors are women. In both Agnini et al. (2020) and Ranganathan et al. (2021), the representation of women faculty is particularly low in geophysics, compared to other geoscience fields. Hori (2020) reported that while women make up 20 % of the Japan Geoscience Union (JpGU) membership, they account for only 2.8 % of JpGU fellows (the JpGU fellowship is an award bestowed upon senior, accomplished academics). While these are distinct snapshots from specific countries, they all show a consistent pattern.

Multiple factors shape this problem. Studies concerning STEM have investigated the effects of social and cultural norms and expectations (Dasgupta and Stout, 2014), implicit bias (Dutt et al., 2016), a lack of role models, science pedagogy (e.g. Hanson, 1996; McGuire et al., 2020), hostile workplace climates for women researchers (e.g. Marín-Spiotta et al., 2020; Casad et al., 2021), choosing to take on larger burdens of care work outside and service work inside academia (Agnini et al., 2020; Ceci and Williams, 2011; Canetto et al., 2012), and even the conception of science itself as a male endeavour due to its historic development (Keller, 2003). A review of various of these arguments can be found in Blickenstaff (2005).

Lerchenmueller and Sorenson (2018) point out that the loss of women researchers in the life sciences does not occur as a steady drip but rather as a heavy spill at critical career junctures, such as the postdoctoral to junior faculty transition. Publication productivity is an important predictor of success during these transitions. underrepresentation of female authors with respect to the presence of women researchers in a research field, as observed by Pico et al. (2020), may consequently be one cause of the continued underrepresentation of women researchers in that field. According to demographic data from the American Geophysical Union (2018)[1], seismology is among the geoscience disciplines with the lowest representation of women. In the present study, we therefore analyse bibliometric data from 14 peer-reviewed journals that are commonly chosen for publishing seismological research in a period of eleven years (2010 – 2020). We build upon the open-source toolkit that Pico et al. (2020) developed to analyse gender in geoscience authorship. This allows us to automatically scrape bibliometric information from journal websites, extract author names, and then obtain their likely perceived gender from requests to web databases relating names to gender. We furthermore propose a method to account for the uncertainty in the author name–gender association by not setting a fixed threshold. For example, Züleyha is considered a female name with 98 % probability, while Hongbo is considered male with 91 % probability, and Andrea is commonly used for both genders (62 % male). By analysing various aspects of the authorship statistics with regard to gender, we aim to document the problem of women's underrepresentation in seismology publications, its recent and possible future development, and point to several consequences that the status quo has both for women seismologists and the field itself. We consider both underrepresentation with respect to the general population

---

[1]Retrieved from: https://honors.agu.org/files/2018/09/2018-section-membership-by-gender-and-career-stage_Sept12.pdf, last accessed 18.08.2022

(assuming a 1:1 gender ratio[2]) and underrepresentation of female authors with respect to the presence of women researchers in the field. To the best of our knowledge, such a detailed study of the authorship gender demographics in seismology has not been undertaken to date, leaving a knowledge gap that needs to be closed in order to support the effort of diversifying all fields of Earth sciences. In our analysis, we specifically focus on the following aspects: i) The overall representation of female authors in seismology; ii) The composition of female/male authors in publication teams; this was examined in terms of author position (first, intermediate, last position in the authors list); iii) The change of female author representation during the past decade; iv) Female author representation per journal; and v) Publication productivity according to the frequency of occurrence of an author in the articles database. Below, we describe our methods and results with regard to these questions, followed by a discussion and conclusions which include our perspective on women's authorship in seismology.

## 2 Method

### 2.1 Collecting bibliometric data

We analyse the representation of female authors in peer-reviewed research articles published in seismology from January 2010 to December 2020. We consider 14 international journals subjectively chosen to cover a broad spectrum of sub-disciplines within the field and a range of impact factors (see Table 1). We collected bibliometric data from the online search masks of the journals, modifying the web-scraping Python code developed by Pico et al. (2020) available online for this purpose. This tool uses the Python package Selenium for opening and downloading search results and the Python package BeautifulSoup to parse the resulting HTML files (Muthukadan, 2022 [last update]; Richardson, 2022 [latest release]). We targeted articles that broadly fall into the field of seismology by selecting all articles with the keyword fragment 'seism' (matching for example seismic, seismological, seismicity) and the keyword 'earthquake' in the abstract. In this way, we obtained entries for 20 108 articles. We extracted the full names of all authors in each article, yielding a list of 88 331 authors. In approximately 20 % of cases, authors chose to use initials rather than first names, and we followed the strategy of Pico et al. (2020) to cross-reference initials and last names with full names in the list of all authors. In addition, we obtained bibliometric information from the SAO/NASA Astrophysics Data System digital library portal (https://ui.adsabs.harvard.edu/) of conference abstracts and used author information of 17 452 abstracts presented at the European Geosciences Union General Assembly during the period 2010 – 2020 to cross-reference initials and last names with full names. In this way, we could identify full names from initials in the majority of cases (> 80 %). Publications with unidentified initialed names were omitted from the database.

---

[2]Human sex ratio is generally not exactly 1:1 for male:female individuals (intersex people do not appear in the statistics we consulted). This is because sex ratio depends on multiple factors such as sex ratio at birth, mortality, and selective abortion (e.g. Ritchie and Roser, 2019). This may also affect gender ratio but how exactly, or whether such data is available, is not known to the authors. Here, we use 1:1 for simplicity.

| Journal | Impact Factor | Number of Articles |
|---|---|---|
| Nature | 42.779 | 59 |
| Science | 41.845 | 78 |
| Nature Geoscience | 14.480 | 169 |
| EPSL | 4.823 | 1239 |
| GRL | 4.50 | 2022 |
| JGR: Solid Earth | 3.64 | 3027 |
| JGR: G3 | 3.28 | 736 |
| SRL | 3.131 | 1452 |
| Tectonophysics | 3.048 | 1606 |
| Solid Earth | 2.921 | 219 |
| GEOPHYSICS | 2.793 | 1753 |
| GJI | 2.574 | 3308 |
| BSSA | 2.274 | 2024 |
| PEPI | 2.237 | 458 |

**Table 1.** Number of articles per journal analysed in this study. We also indicate the 2-year impact factor reported by each journal in 2021. EPSL: Earth and Planetary Science Letters; GRL: Geophysical Research Letters; JGR: Journal of Geophysical Research; G3: Geochemistry, Geophysics, Geosystems; SRL: Seismological Research Letters; GJI: Geophysical Journal International; BSSA: Bulletin of the Seismological Society of America; PEPI: Physics of the Earth and Planetary Interiors.

## 2.2 Relating first names to gender

We use several web databases to infer author gender from the name. We first submitted all first names in the database to the genderize.io API (https://genderize.io/) used by Pico et al. (2020). For each name, we stored the likely gender of the name ("female", "male" or None) and the probability of the gender returned by the API. Using genderize.io, we identified 73 % likely male, 20 % likely female, and 7 % not classifiable names. For names that could not be classified (None result), we repeated the process with the NamSor API (https://github.com/namsor/namsor-python-sdk2), which requires first and last name as input, and uses public "labelled" data such as voter registration lists, but also linguistic cues, such as name endings. This resulted in a combined identification (from both genderize.io and NamSor) of 76 % likely male, 20 % likely female, and less than 4 % unclassified author names. After removing articles with any unclassified author names, our final dataset contains 18 150 articles (Table 1).

## 2.3 Representing gender through probabilities

The online databases to determine the gender of author names return a probability that the name in question is male or female. As is common practice, Pico et al. (2020) set a threshold at 0.5 to distinguish male and female names. In contrast, we retain

the probability returned by the online tools and base our analyses on it. Using a fixed threshold can distort the results because not all first names are unambiguously gendered. As an example, consider the name Ashley, which is classified as female by

genderize.io, but with a probability of only about 0.6. If our dataset contained 10 authors named Ashley, and we used a cut-off to assign them a binary gender, 10 out of 10 would be considered female. By continuing to work with the probability instead, and interpreting probability in terms of frequency, 6 out of 10 would be considered female, and 4 male, which provides a more accurate picture of the demographics.

### 2.3.1   Computing probabilities

Here, we summarize the most relevant mathematical operations used for computing the probabilities in Sect. 3. For our analysis, we consider a binary notion of name gender so that the probabilities of male and female add up to 1. Our database contains a total of $n$ articles, and we denote an article as $x_i$ with $i \in \{1, ..., n\}$. From the output of the gender determination tools, we obtain the conditional probability $p(F_k \mid x_i)$ of having a female-gendered author name at the authorship position $k$ in the article $x_i$. For example, $p(F_1 \mid x_i)$ refers to the probability of having a female-gendered first-author name in the article $x_i$. Then, the

overall probability of having a female-gendered first-author name in our database can be computed as

$$p(F_1) = \sum_{i=1}^{n} p(F_1 \mid x_i) p(x_i). \tag{1}$$

We consider all articles equally likely; thus, $p(x_i) = 1/n$ for all $i$, so that Eq. (1) reduces to the arithmetic mean of $p(F_1 \mid x_i)$. Similarly, the overall probability of having a male-gendered first-author name is given by

$$p(M_1) = \sum_{i=1}^{n} \left(1 - p(F_1 \mid x_i)\right) p(x_i) = 1 - \frac{1}{n} \sum_{i=1}^{n} p(F_1 \mid x_i) = 1 - p(F_1). \tag{2}$$

We use Eqs. (1) and (2) indistinctly for any authorship position by replacing $p(F_1 \mid x_i)$ with the appropriate probability. For last-authors this becomes $p(F_{\text{last}} \mid x_i)$, and for co-authors we use

$$p(F_{\text{co}} \mid x_i) = \frac{1}{m-2} \sum_{k=2}^{m-1} p(F_k \mid x_i), \tag{3}$$

where $m$ is the total number of authors of the article, and we assume uniform distribution for co-authorship positions. For the sake of this analysis, we define co-author as any author that is neither first nor last.

To analyse the gender composition of author teams, we start by computing the probability of an author list with only same-gender author names. For instance, the probability of all author names being female in the article $x_i$ is computed as

$$p_i(F_{\text{all}}) = \prod_{k=1}^{m} p_i(F_k), \tag{4}$$

where $p_i(\cdot) := p(\cdot \mid x_i)$, and we assume that the genders of the authors are independent within each individual article. We follow the same procedure to compute the probability of all author names being male $p_i(M_{\text{all}})$. Finally, the probability of having a

mixed-gender author names list can be derived as

$$p_i(\text{mix}) = 1 - p_i(F_{\text{all}}) - p_i(M_{\text{all}}). \tag{5}$$

We could also estimate the probability of having at least one female-gendered author name in the author list by computing $p_i(F_{\text{at least one}}) = 1 - p_i(M_{\text{all}})$. Similar to Eqs. (1) and (2), we use the arithmetic mean to estimate the overall probabilities of $p(\text{mix})$, $p(F_{\text{all}})$, and $p(M_{\text{all}})$.

To investigate gender dynamics in the composition of author teams, we derive conditional probabilities of the first-author gender given the last-author gender and vice versa. For instance, we compute the probability of having a female-gendered first-author name given that the last author name is also female-gendered as

$$p(F_1 \mid F_{\text{last}}) = \frac{p(F_1 \cap F_{\text{last}})}{p(F_{\text{last}})} = \frac{\sum_{i=1}^{n} p_i(F_1 \cap F_{\text{last}})}{\sum_{i=1}^{n} p_i(F_{\text{last}})} = \frac{\sum_{i=1}^{n} p_i(F_1) p_i(F_{\text{last}})}{\sum_{i=1}^{n} p_i(F_{\text{last}})}. \tag{6}$$

Note that the assumption of independence between first and last author of a single article in the last step of (6) does not
imply that the overall probability of having a female-gendered first-author name is independent of the last author gender, i.e., $p(F_1 \mid F_{\text{last}}) = p(F_1)$. Equations similar to (6) can be defined for different combinations of first- and last-author genders or for different probabilities of interest (e.g., probability of having at least one female co-author given that the last author is female).

## 2.4   Statistical analysis

Statistical analysis and visualization were performed using Python (version 3.8.10) with the SciPy (1.7.1), Pandas (1.3.2),
and Seaborn (0.11.2) libraries. We used Pearson's $r$ coefficient to calculate correlations of probabilities with publication years and Spearman's $\rho$ coefficient for correlations with journal impact factors and the number of authors. Results were considered significant for $p$-values lower than 0.05. We used the slope of linear regressions to analyse the increase rate per year of the probabilities and forecast when the parity is reached. Additionally, we provide the average annual growth rates (AAGR) of the probabilities for direct comparisons with similar studies.

# 3   Results

## 3.1   Overall representation of female authors

The overall representation of female-gendered author names in our database is 23.6 %, approximately one-fourth of the total number of authors of the publications analysed, and more than the frequency of names classified as female when using a threshold of 0.5 for determining the gender (20.6 %). As shown in Fig. 1a, they appear most likely in the first authorship
position (with a probability of 0.28), followed by the co-authorship (0.23) and the last authorship position (0.19). Seven (eight) out of ten articles therefore have a male-gendered first-author (last-author) name.

## 3.2   Gender composition of author teams

The percentage of articles with all female-gendered author names is only 2.7 % (Fig. 1b). That is, male-gendered author names appear in 97.3 % of publications. In contrast, 41.3 % of articles do not contain any female-gendered author name. Male authors
are therefore 15 times more likely to work in same-gender teams than the female authors. Publications with mixed-gender author lists are most common in seismology, but not yet the norm (56.0 %).

If authors in a team are chosen at random, larger author teams are expected to be more diverse in terms of gender. We indeed find that the probability of having at least one female-gendered author name is significantly and positively correlated with the number of authors of a publication ($\rho = 0.52, p < 0.0001$). On average, it increases from 0.17 for single-authored articles to 0.93 for those with twelve authors (Fig. 2a). However, when comparing these observations with the expected values computed from the overall female author representation (0.24) assuming randomly composed teams, we find a negative bias regardless of the team size (Fig. 2b). Note that for articles with more than seven authors, the sample sizes are below 5% of the total number of articles; thus, observed biases in the case of large teams should be carefully interpreted due to the reduced statistical significance. Single-authored publications show the strongest underrepresentation of female-gendered author names, with a negative bias of 6.5 % compared to their expected representation.

Based on the average probabilities, a female-gendered author name has a 95 % probability of appearing in a publication when the number of authors is twelve, whereas for male-gendered author names the number of authors required to reach the same probability is three. In other words, a trivial consequence of women authors being a minority is that only large author teams make their presence highly likely, and only very large teams make it likely that they are not the only one of their gender. The median number of authors in our database is four, corresponding to a 0.66 expected probability of at least one female-gendered author name appearing in any position, while the observed value for this probability is 0.59 (Fig. 1b).

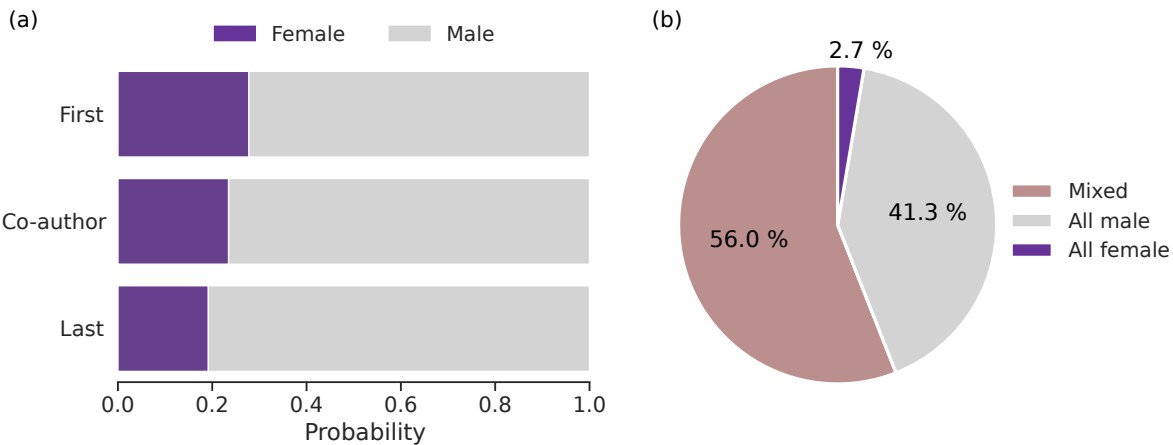

**Figure 1.** Gender distribution in the authorship list of peer-reviewed publications in seismology. (a) The probability of having a female- or male-gendered first-author, co-author, and last-author name in a publication. (b) Percentage of publications with an authorship list with all-female, all-male, or mixed-gender author names.

### 3.3 Composition of author teams conditioned on first- and last-author gender

As Fig. 1b shows, working in mixed-gender author teams is the norm for female authors, while there is a substantial probability for male authors to work in all-male author teams. To investigate further whether the author team composition can be considered

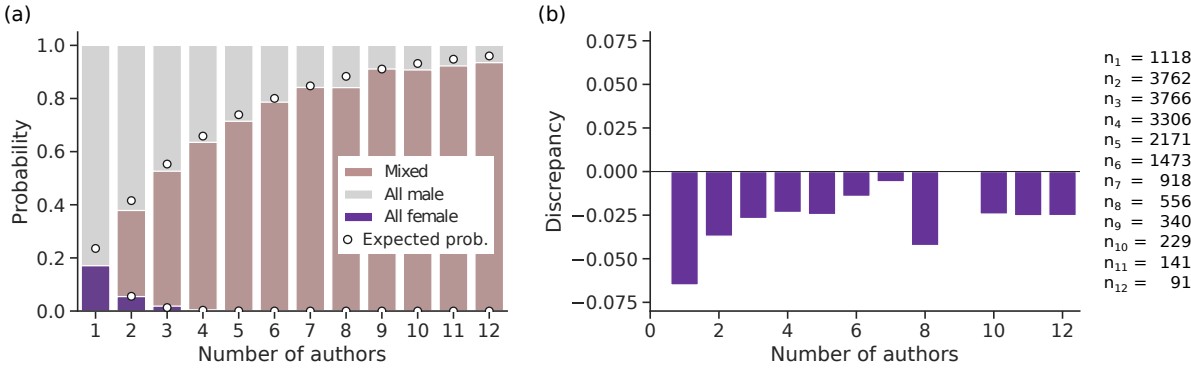

**Figure 2.** Probabilities with respect to the number of authors of a publication. (a) Expected and observed probability of an all-female, all-male, or mixed-gender author list. (b) Discrepancy between the observed probability of having at least one female-gendered author name and the expected probability for randomly composed teams. The sample size for each number of authors is indicated as $n_{\text{number of authors}}$.

random with respect to gender, we evaluate the probabilities of the first- and co-author gender conditioned on the last-author gender and probabilities of the last- and co-author gender conditioned on the first-author gender. The results in Fig. 3a suggest that female-gendered first (last) author names are 4.4 % (3.4 %) more likely when the last (first) author name is also female-gendered, displaying a slight "gender unmixing" effect in teams consistent with the bias observed in the previous section. We observe a similar effect in the probability of having at least one female-gendered co-author name, which is 6.4 % larger for female last authors than male last authors. We find that female first authors are 16.8 % less likely than male first authors to work with female co-authors.

### 3.4    Changes in the last decade

As shown in Fig. 4a, probabilities of having female-gendered names in first-, co-, and last-authorship positions have increased over the last decade (first author: $r = 0.81, p = 0.002$; co-author: $r = 0.95, p = 1 \cdot 10^{-5}$; last-author: $r = 0.85, p = 0.001$). The increase rate per year is fastest for co-authors ($0.6 \pm 0.1$ % ; AAGR: 2.4 %), twice that of the first and last authors ($0.3 \pm 0.1$ %; AAGR: 1.5 % and 1.7 %, respectively). Assuming these annual rates are constant in the following years, a probability of 0.5 representing parity with respect to the general population will be reached in $42 \pm 5$, $72 \pm 17$, and $93 \pm 20$ years, respectively, in the co-, first, and last author position.

The composition of author teams is rapidly becoming more mixed in gender with time ($r = 0.97, p = 5 \cdot 10^{-7}$). In 2020, publications with mixed-gender authors were approximately 13 % more likely than in 2010 (Fig. 4b). Consequently, there has been a substantial drop in all-male-authored publications, from 48.2 % in 2010 to 36.0 % in 2020. The mean number of authors of a publication has also increased by almost one author in the last decade (Supplementary Fig. S1), a trend that is also observed in other fields of scientific publishing (e.g. Kuld and O'Hagan, 2018, in economics). This may have contributed to

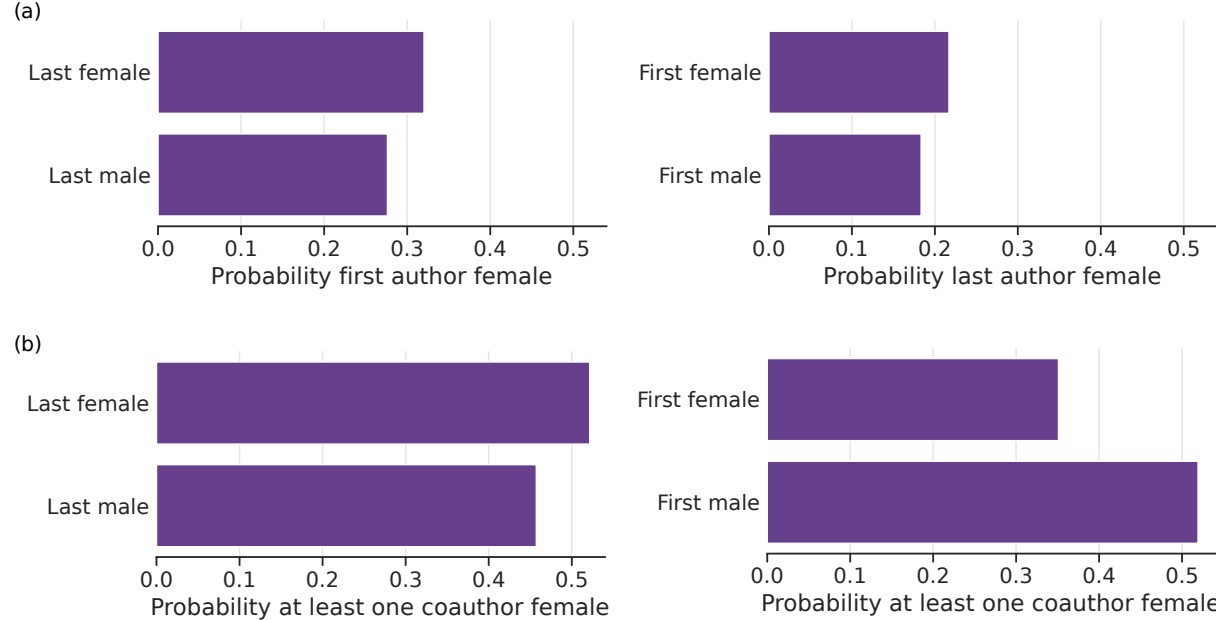

**Figure 3.** Conditional probabilities analysing author team composition. (a) Probability of having female-gendered first-author names (left) and last-author names (right), given the gender of the last and first author, respectively. (b) Probability of having at least one female-gendered co-author name given the gender of the last (left) and first (right) author.

diversifying author teams, as already discussed in Fig. 2. With the observed annual rate, approximately all publications will be authored by mixed-gender teams in $26 \pm 1$ years.

### 3.5 Representation by journal

Figure 5 shows the probabilities of having female-gendered names in first-, co-, and last-authorship positions for each journal. We group the journals Nature, Science, and Nature Geoscience (Nat/Sci) to increase robustness, as our dataset contains a relatively small number of publications in these journals ($n = 306$). In general, by-journal probabilities are close to the overall probabilities shown in Fig. 1a. However, three bins stand out: (i) The journal GEOPHYSICS has the lowest female representation in all authorship positions. Compared to the overall probabilities, the underrepresentation is most substantial for first authors (negative bias of 7 % compared to the average appearance of female-gendered author names), followed by last authors (5 %) and co-authors (3 %). This journal also shows the smallest mean number of authors per publication ($3.3 \pm 1.5$, Supplementary Fig. 2). (ii) The journal Geophysics, Geochemistry, Geosystems (G3) has an elevated probability of female-gendered first author names (positive bias of 6 %). (iii) The journals Nat/Sci, which have the highest impact factors, have a lower than overall female representation among first authors (negative bias of 5 %) and last authors (3 %). Female-gendered author names are therefore more likely to appear in the co-authorship position, unlike the general trend observed in Fig. 1a. These journals show the largest mean number of authors per publication ($6.6 \pm 7.4$, Supplementary Fig. S2). Despite high-impact journals hav-

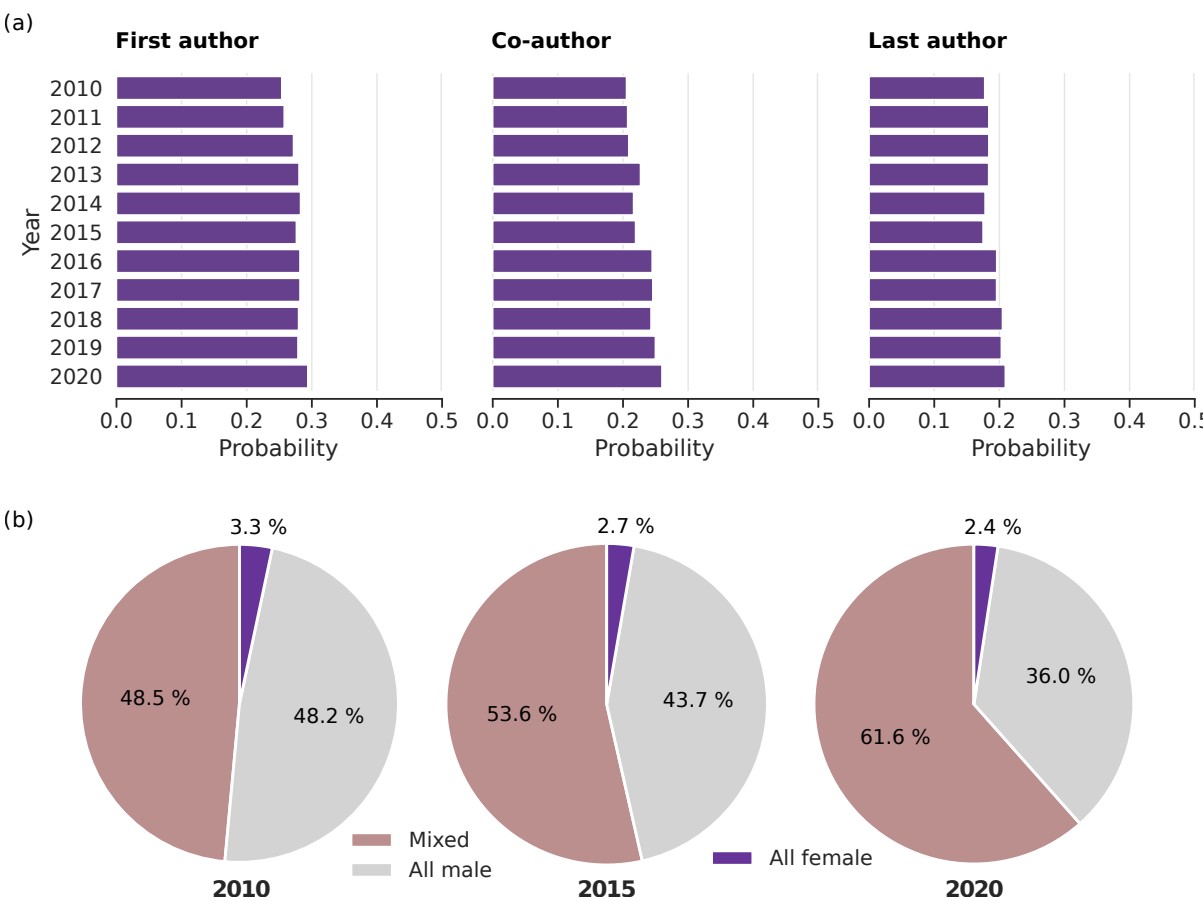

**Figure 4.** Changes in female author representation and gender composition of author teams during the last eleven years. (a) Probabilities of first-author (left), co-author (center), and last-author (right) names being female-gendered per year in articles published from 2010 to 2020. (b) Percentage of publications with an authorship list with all-female, all-male, and mixed-gender author names in 2010 (left), 2015 (center), and 2020 (right).

ing a significantly larger number of authors per publication ($\rho = 0.71, p = 0.004$), we do not observe any correlation between
the probability of having at least one female-gendered author and the journal impact factor ($p = 0.3$).

### 3.6 Towards comparing demographic and bibliographic data

Since 2016, the European Geosciences Union (EGU) systematically collects self-declared demographic information from participants upon membership registration. In 2016–2017, response rates to the question about gender were low, around 50 % for overall seismology section attendees and around 40 % for early career scientists (ECS). The response rates increased in
2018–2019 to around 60 % (ECS: around 50 %) before reaching close to 100 % in 2020.

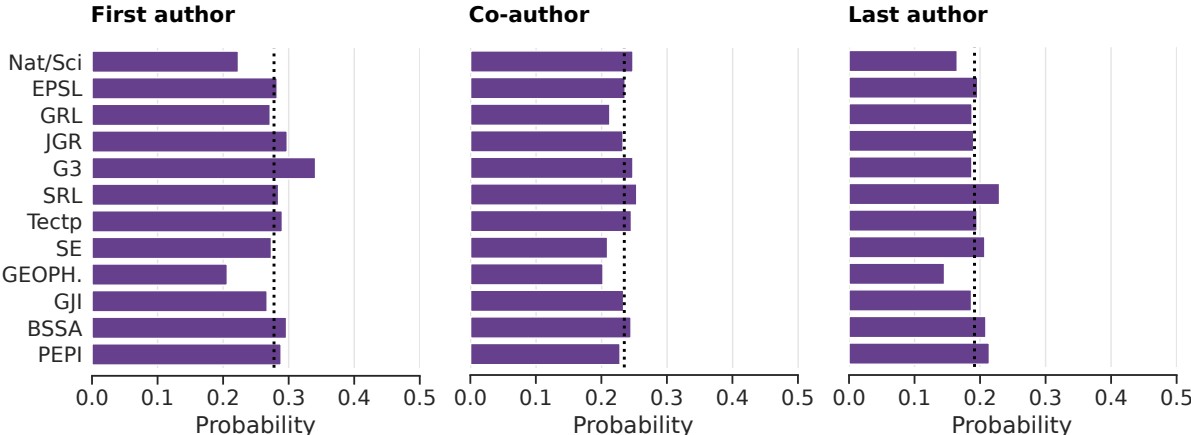

**Figure 5.** Probabilities of first-author (left), co-author (center), and last-author (right) names being female-gendered per journal. Dashed lines indicate the overall probabilities from Fig. 1a in each case. Nat/Sci: Nature, Science and Nature Geoscience; EPSL: Earth and Planetary Science Letters; GRL: Geophysical Research Letters; JGR: Journal of Geophysical Research: Solid Earth; G3: Geochemistry, Geophysics, Geosystems; SRL: Seismological Research Letters; Tectp: Tectonophysics; SE: Solid Earth; GEOPH.: GEOPHYSICS; GJI: Geophysical Journal International; BSSA: Bulletin of the Seismological Society of America; PEPI: Physics of the Earth and Planetary Interiors.

Between 2018 and 2021, 29–33 % of all seismology section members and 35–38 % of the early career members identified as women. For the years with higher response rates, and for both levels of seniority, EGU seismology members have a larger probability of being female than manuscript authors in our dataset. The EGU demographics refer to unique members of the seismology community, while our overall probabilities are computed for authorships, i.e. one person may appear repeatedly. The discrepancy in women's participation might indicate a gender gap in authorship. However, several points prevent a direct comparison of both datasets: (i) The results of automatic genderization are less reliable than self-declaration; (ii) While we expect a large overlap of the researchers represented in both datasets, section members may be more frequently working in European countries than article authors; (iii) Our dataset can distinguish first authors (who are often, but not necessarily early-career researchers) from other authors, while EGU considers self-declared ECS; these two populations are not directly comparable. Considering these limitations, we merely conclude that there are several consistent points between our results and the section membership data, namely that ECS members / first authors are more likely to be women than overall members / authors, and that the rate of women membership / authorship up to 2020 is between 20 and 30 % for all members / authors. We cannot conclusively state that a gender gap in publication productivity exists due to the mentioned caveats, but based on our results, such a gap is possible.

## 3.7 Publication productivity

As we have seen above, it is difficult to compare the populations of conference participants (with possible geographic preference) to article authors. Therefore, we take a second approach to analyse the effect of gender on productivity using only our

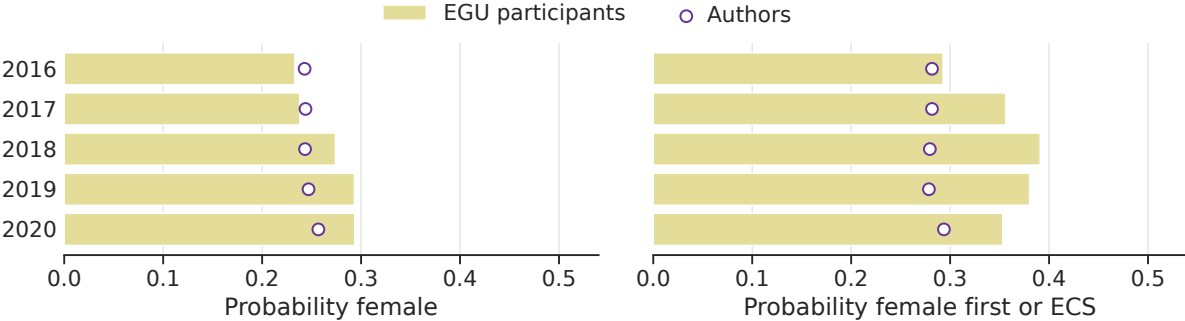

**Figure 6.** Comparison of probability of female authorship in our article database to demographic data of conference participation at the European Geosciences Union General Assembly. Left panel: all authors. Right panel: first authors of articles and early career scientist participants at EGU.

data set. We identify unique full names of first and last authors in the database and rank them by the number of their publications. Fig. 7 (purple curve) shows how the probability of female-gendered first and last author names evolves with the ranking by productivity. Considering all 9305 unique first authors in the first ten years of the database, the probability of unique female-gendered author names among them is close to 0.3. However, when we consider only the most productive 10 % of authors, this drops to 0.25. Among the most productive few percent, probability appears to drop further; however, the absolute number of female-gendered first-author names is too low in this range to draw statistically meaningful conclusions. Last author names ranked by productivity show the same pattern. Hypothesizing that this lack of highly productive female authors is an effect of women entering the field later, we separate the data for the years 2010-2014 and 2015-2019 (blue and pink curves in Fig. 7, respectively). We can make several observations based on these results. (i) The number of unique authors has increased by 21 % in the span of five years. (ii) Authors with female-gendered names are overall more strongly represented in 2015-2019. (iii) This increase in overall representation includes some of the most productive authors ranked between 10 % and 20 %, but a gender gap among highly productive authors still persists.

## 4   Discussion

We present the first study analysing the representation of women researchers in peer-reviewed articles in the field of seismology. In geosciences, of which seismology is a sub-discipline, the gender distribution of first-author names has already been analysed by Pico et al. (2020). Here, we focus on a subfield and extend the scope to analyse women's representation in first-, co-, and last-authorship positions, as well as gender dynamics in the author team composition and gender differences in publication productivity. In addition, we use a new approach to estimate the probability of author genders. Rather than considering a threshold to assign a binary gender to each name (either female or male), we use the probability of names being female-gendered. In this way, we automatically account for uncertainties in name-gender association, with no need to disregard names that are not gender-specific (e.g., Bendels et al., 2018; Pico et al., 2020).

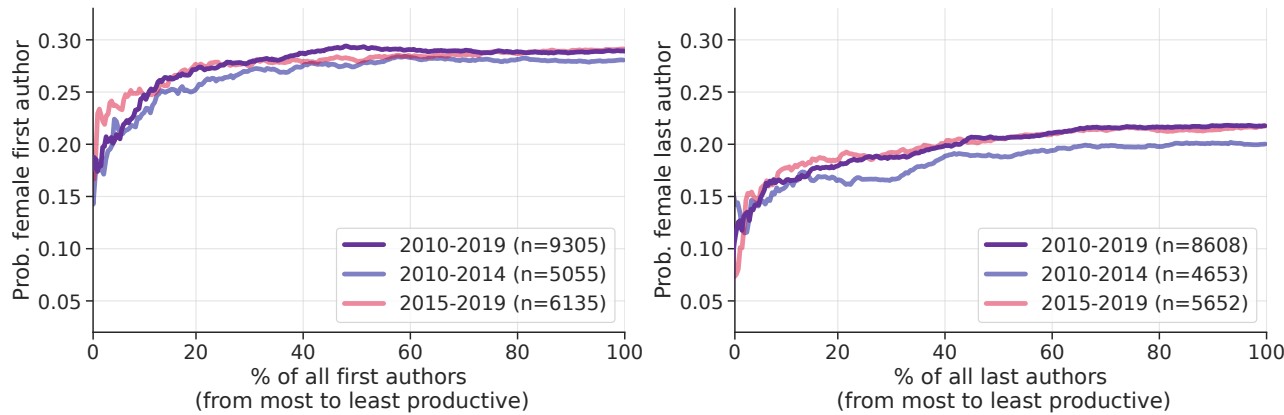

**Figure 7.** Female author representation by ranking in terms of output. We show the probability of female first and last author names as a function of productivity quantile from most to least productive, where productivity is measured as the number of articles published. We do not interpret the most productive 10 % (small number of authors).

Throughout this discussion, we compare our findings to the literature on gender gaps in authorship in STEM and other fields of research. Although direct comparisons to other academic fields could be simplifying, they provide an indication of the general trend of women's underrepresentation in authorship and its subsequent effects in academic career progression. Moreover, they allow us to illustrate some of the possible consequences of authorship gender gaps that have not been studied in geosciences yet. The comparisons should be read as a motivation to conduct further studies that aim to address and quantify gender inequities in the field of seismology or geophysics, rather than as a quantitative discussion.

In this study, we found that women researchers in seismology account for 24 % of authorships, precisely the same rate reported in earth and environmental sciences by Bendels et al. (2018). For every ten publications in seismology, only three are authored by female first authors and two by female last authors. Thus, women researchers are most likely to publish in seismology as main contributors and least likely as senior authors or project leaders, who tend to be named last. The overall female author odds ratios, which indicate how female authorship is split between first, co- and last author positions, are 1.4, 1.0, and 0.7 for first, co-, and last authors in our dataset, while they are approximately 1.5, 0.9, and 0.7 in Bendels et al. (2018). This difference in representation among authorship positions evidences the progressive absence of women in academic positions with higher responsibility, commonly described as the leaky pipeline (e.g., Blickenstaff, 2005). In terms of unique authors, we found that these are female with a probability of 26% (any author position), 29% (first author position), and 22% (last author position). These numbers correlate well with the proportion of unique active women authors of peer-reviewed publications in natural sciences compiled by the She Figures report based on global bibliographic data from Scopus (European Commission, Directorate-General for Research and Innovation, 2021, p. 221). Among these active authors, there are 29% of women in the early to mid-career segment and 23% of women in the established segment. Although our study considers a specific subfield of natural sciences and a different genderizing strategy, the comparison provides an important indicator that our automated

procedure of retrieving and genderizing author names is reliable because it produces results close to those of a larger, more general dataset.

Similar to women researchers in other disciplines, women seismologists are particularly under-represented in single-authored publications (West et al., 2013; Walker, 2019). Sarsons (2017) found that, in economics, this type of underrepresentation penalizes female researchers' odds of receiving tenure while playing no significant role in the promotion of male researchers. In seismology, solo articles are only a small fraction of all articles (approx. 6 %), with an overall declining trend observed in multiple fields of research (Kuld and O'Hagan, 2018; West et al., 2013). However, this gap may still affect the career advancement of women researchers, as single-authored papers are considered proof of the authors' scientific skills (Sarsons, 2017; Kwiek and Roszka, 2022).

The vast majority of publications authored by women seismologists (95 %) are submitted by mixed-gender author teams, meaning that women routinely collaborate with colleagues diverse in gender. In contrast, male researchers co-publish exclusively with other male-gendered colleagues once every three articles. This is not only caused by seismology being a male-dominated field but goes beyond what we would expect from a random team composition (Fig. 2). Similar to what has been reported in other STEM fields (Ghiasi et al., 2016; Holman and Morandin, 2019), researchers in seismology appear to work with same-gendered co-authors more often than expected. However, this effect is not necessarily mutual: as will be further discussed below, it is possible that only one gender displays a same-gender preference. When the majority group displays this preference, it constrains the possibilities for the minority group to participate in academic networking and access novel ideas, information, and research opportunities (e.g., McPherson et al., 2001).

We furthermore observe that the probability of women first authors increases in publications with a woman last author, and vice versa (Fig. 3a). Studies on unconscious bias suggest that this is not driven by the hiring preferences of women faculty, which tend to be the same as those of male faculty (Moss-Racusin et al., 2012). We hypothesize that the "unmixing" may in part be a geographic effect, i.e., that women researchers are not randomly distributed across universities, but tend to be better represented in more equitable institutes and countries, and collaborate more frequently within their institute and within their country (Kwiek and Roszka, 2021; Hanson et al., 2020). Another contributing factor may also be the implicit gender bias of male researchers. Araújo et al. (2017) found that women researchers are equitable in their collaborations, with their collaborator pools representing the gender balance of their fields, but male researchers show a preference for working with other males. Similarly, Hanson et al. (2020) demonstrated that while female AGU abstract authors have a higher rate of connections to other women compared to male abstract authors, the female-female rate of connections reflects the actual share of women among AGU members, whereas the male-female rate of connections remains below the share of AGU women members. If we assume that EGU demographic data (Fig. 6) accurately represents the distribution of women researchers in the field, our results indicate that women's underrepresentation in important authorship positions is indeed stronger with male first or last authors.

A curious finding in this context is related to the conditional probability that at least one co-author name is female-gendered given a last or first female-gendered author name (Fig. 3b). It is comparatively high when the last author name is female-gendered but comparatively low when the first author name is female-gendered. It shows that: (1) Women senior researchers have a positive effect on overall participation of women in seismology authorship, similar to the productivity increase shown

by women graduate students working with female advisors (Pezzoni et al., 2016). (2) Early-career women seismologists tend to be more isolated from women co-authors who might be either additional senior researchers or peers at a similar career stage. Although by the time of this work, we did not find studies supporting or contradicting this observation, it agrees with our own experience in the field.

Women's authorship has been slightly increasing from 2010 to 2020 for all authorship positions, most strongly in the co-author position. Both our observations and the self-reported demographic data of the EGU seismology section members indicate that women's participation does not increase at a steady rate, but fluctuates year by year. We have used a linear model to fit the increase in the probability of women's authorship with time. Conversely, it is common practice to assume an exponential model and report growth in women researchers' participation in terms of compound or average annual growth rates (Bendels et al., 2018; Pico et al., 2020; European Commission, Directorate-General for Research and Innovation, 2021). Obviously, the outlook changes drastically when an exponential vs. a linear model is used. Besides the linear increase rates of female author probabilities (0.3, 0.6, and 0.3 % / year for first, co-, and last authors), we find compound annual growth rates of 1.5, 2.4, and 1.7 % for the probability of female first, co-, and last authors, respectively. The rates we find for first and last authorship are roughly in agreement with the overall rate in natural sciences in the European Union determined by the European Commission (1.35 %) for the years 2015 – 2019, while the co-author rate is well above it (European Commission, Directorate-General for Research and Innovation, 2021). Our rates fit less well with Bendels et al. (2018), who report corresponding average annual growth rates of approximately 3, 1.5, and 2 % in Earth & Environmental sciences (Fig. 2 in Bendels et al., 2018), which is likely caused by a different choice of analysed journals.

As to the future development, there is little indication of exponential growth of women's authorship in our 11-year dataset. The compound annual growth rates determined from the data are low enough for the exponential model to behave almost linearly (Supplementary Fig. S3). A model determined by linear regression actually provides a better fit to the data in all cases, especially for first authorship. Ultimately, our observation period is too short to answer this question, particularly in light of the strong year-to-year fluctuations. Longer-range data are needed to study the development in more detail, make better-founded predictions, and investigate which factors affect temporal fluctuations in diversity.

Female-gendered first names are less likely to appear in the first and last authorship position in high-impact seismology articles than in most other journals, with the exception of GEOPHYSICS. First authorship in high-impact journals can be a career-advancing achievement: despite efforts to abolish the journal impact factor in hiring and promotion decisions (e.g. Lariviere and Sugimoto, 2019), using it as a performance metric has been commonplace and continues at many institutions (McKiernan et al., 2019). Thus, publishing in high-impact journals at lower rates may put women seismologists at lower odds of being hired and promoted.

The most highly productive authors are more likely to have male first names than authors of average productivity, both for first and last authors. This gap has been slightly narrowing over time, but still persists. A productivity gender gap has been documented in various areas of STEM (Larivière et al., 2013; Bravo-Hermsdorff et al., 2019; Lerchenmueller and Sorenson, 2018; Bendels et al., 2018; European Commission, Directorate-General for Research and Innovation, 2021, and references therein). A recent study suggests that women researchers are less likely to be offered authorship in collaborative projects

(Ross et al., 2022), leading to an "attribution gap" that could explain parts of the productivity gap. Whatever its causes, it likely presents an obstacle to increasing the participation of women in seismology and reaching parity at the faculty level. Lerchenmueller and Sorenson (2018) found that the lower success rate of women in life sciences during the transition from postdoctoral to principal investigators (PI) is to 60 % explained by their lower overall productivity and that the remaining gap can be almost entirely accounted for by the effect that outstandingly productive authors are more highly rewarded (so-called Matthew effect).

The results of our study are limited to the small sample of selected journals. They have been chosen according to their impact factor and popularity based on our experience as seismologists in European institutions. We include Nature, Science and Nature Geoscience because of their prestige. The remaining 11 journals are all in the first quartile of journals in geophysics in terms of their number of citations and articles published in 2018-2021 (Scopus). Furthermore, this study does not use self-declared gender data because the journals we consider do not collect and publish them. Instead, we use the term 'gender' to describe the likely perceived gender based on author names using publicly available name-to-gender inference tools (i.e., genderize.io and NamSor). While these tools are frequently used in similar studies (e.g., Pico et al., 2020; European Commission, Directorate-General for Research and Innovation, 2021), they have two main limitations: (1) They assign a binary gender (female/male) and a probability to each name. By explicitly using these probabilities, we intend to consider gender identity as a continuous spectrum where female/male appear at the two extremes. However, this does not necessarily represent the true complexity and multiplicity of gender identities. (2) They assume that the binary gender of a person can be inferred from Latin transcriptions of their first (genderize.io) or full names (Namsor). Santamaría and Mihaljević (2018) reported overall inaccuracies of approximately 5% when assessing the performance of genderize.io and NamSor against manually annotated author-gender datasets, with particularly poor performance for Asian names ($\sim$12% of inaccuracies). Since both gender-inference tools assign systematically low probabilities to Asian names (Santamaría and Mihaljević, 2018), considering gender probabilities rather than fixed thresholds reduces their contribution to our results. This means that misclassifications affect our results less than they would if we were using a fixed threshold. However, it also means that our results reflect the gender distribution of authors with non-Asian names more accurately than that of authors with Asian names. Finally, our study does not consider equally contributing authors and assumes that the seniority of a researcher is reflected by their position in the author list, with the last author being more senior than the first one. This is common in seismology, but may not be applicable in other scientific fields.

## 5 Conclusion

By analysing bibliographic data, we identified several gender gaps in seismology authorship. Firstly, the probability of first authorship of women in recent years is below the probability of women Early Career Scientists conference attendance at the EGU general assembly. Secondly, the probability of having at least one woman author team member is below what would be expected if teams were assembled randomly, which points to a gender bias in how authorship teams are composed. Thirdly, female-gendered first names are less represented among the most productive authors, the first and last authors in prestigious high-impact journals, and solo authors than they are overall in our dataset. Women's representation as authors in seismology

has improved from 2010 to 2020, but the improvement is fastest for the less prestigious co-author positions and is moving at a rather slow and stumbling pace. Given the observed rates, we can estimate that:

- current early career researchers in seismology would experience that author teams become universally mixed-gender towards the end of their career;

- undergraduate students entering university in 2022 and going on to become seismologists would experience gender parity in seismology co-authorship;

- people born in the early 2020s who decide to become seismologists might experience parity in first authorship in seismology but would probably spend their entire careers still waiting for parity in last authorship.

A more optimistic scenario is possible if gender diversity increases exponentially as Pico et al. (2020) and Bendels et al. (2018) assume (instead of 42, 72 and 93, it would take approximately 30, 40 and 50 more years to reach gender parity in seismology co-, first and last authorship). Based on our findings, we can offer the following comments and recommendations to actively work towards gender diversity in seismology authorship:

- Our analysis profited from comparison to carefully collected and curated demographic data, e.g., by the European Geosciences Union. We encourage professional societies to collect or continue collecting demographic data where appropriate and where sufficient anonymity can be granted. As representation in professional societies and conference attendance are not equal to publication productivity, journal publishers should also consider collecting inclusive and self-declared demographic data.

- Those evaluating research performance should remain aware that there are, as of now, gender gaps in high-productivity, solo, and high-impact authorship in seismology. If this is not taken into consideration in funding and hiring decisions, it may contribute to perpetuating the leaky pipeline problem.

- To understand the root causes of these gaps, an effort in the form of continued studies should be made to understand why there is a high-productivity, solo-author, and high-impact gender gap in seismology publications. This should ideally be based on self-declared inclusive demographic data of authors (e.g., Strauss et al., 2020).

- Research groups and their PIs should engage in transparent and ongoing communication about authorship criteria. In addition,they should consider establishing criteria defining what output merits submission to a high-impact journal.

- PIs should pay particular attention to the opportunities for collaboration that they can offer to their female (and, by proxy, all minority) mentees. If the gender "unmixing" is not a geographic effect, it means that these individuals have fewer choices for collaboration.

- For the case that the "unmixing" effect is geographic, researchers who cannot find women collaborators at their institute may support diversity by seeking them elsewhere and internationally.

Importantly, we have only considered the influence of gender on seismology authorship. Specific to context, researchers can hold various, sometimes intersecting minority identities who face serious obstacles in the geosciences (e.g., due to racism; Bernard and Cooperdock, 2018; Dutt, 2020; Dowey et al., 2021). In this work, we have focused on women's underrepresentation in authorship of seismology papers because we are part of this minority. However, we sincerely hope that the seismological community will continue to uncover and remove obstacles for all minority researchers and work towards becoming more diverse and inclusive.

## 6 Code availability

All the tools we used in this research are openly available and can be obtained through GitHub or the Python Package index as indicated by URLs and references.

## 7 Data availability

All data used in this study is publicly available. For the convenience of anyone who wishes to use it, the dataset of bibliographic information will be provided after appropriate anonymisation (i.e., without author names and article titles) by the corresponding authors upon request.

## 8 Author contributions

All authors contributed equally in all steps involved in this study and manuscript writing.

## 9 Competing interests

The authors declare that they have no conflict of interest.

## 10 Acknowledgements

We are grateful to an anonymous reviewer and Ben Fernando for constructive comments, as well as the editor, Caroline Beghein, for handling the manuscript. We thank Elian Carsenat from NamSor for supporting the research by providing us with credits to genderise our author database and for a helpful discussion on literature about genderising algorithms. We would also like to thank the European Geosciences Union, in particular Dr. Philippe Courtial, for collecting and making accessible membership demographic data.

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
