# Peer review of "Quantifying gender gaps in seismology authorship"

_EGUsphere, 2022_

## Author Comment (AC1)

**Referee #1: Anonymous**

*Paper investigates female authorship of peer-reviewed publications in seismology. The topic of the paper is of high significance as authorship of scientific peer-reviewed papers remains an important criterion for assessing researchers' performance and consequent career advancement. Therefore, any biases or underrepresentation of any demographic groups may lead to lower chances of recognition (job opportunities, career progressions, funding, etc.).*

*I will not comment on the probabilistic approach of determining the gender of authors based on the first name nor on the statistical method. However, the size of the sample used is sound in terms of statistical significance. The overall reasoning and justifications of various decisions is very appropriate.*

*The results are quite relevant and of interest. The representation by journal is very useful and (potentially) an eye-opener to both female and male authors.*

*Q 1.1 In the discussion, the authors compared their results to those of the European Commission. They observed a correlation between 24% of authorship by women and 30% women representation in natural sciences. I am not sure it adds much to the discussion as this comparison is very difficult and thus any correlation is oversimplified.*

We thank the reviewer for pointing out this sentence. Actually, it was poorly formulated: The observations of Lariviere et al. (2013) and also the "She Figures" report from the European Commission refer to women authorship, not to representation in general. However, going back to the original data, we also noticed that the measures we were comparing were not entirely comparable (we had been comparing overall women's authorship in our data to the average proportion of women authors *per article* in the She Figures report). We now changed this part of the discussion so as to include a slightly more detailed comparison to the values in the She Figures report and a rationale for performing this comparison.

Added text in lines 276 – 283:

"In terms of unique authors, we found that these are women with a probability of 26% (any author position), 29% (first author position), and 22% (last author position). These numbers correlate well with the proportion of unique active women authors of peer-reviewed publications in natural sciences compiled by the She Figures report based on bibliographic data from Scopus (European

Commission, Directorate-General for Research and Innovation, 2021, p. 221). Among these active authors, there are 29% of women in the early to mid-career segment and 23% of women in the established segment. Although our study considers a specific subfield of natural sciences and a different genderizing strategy, the comparison provides an important indicator that our automated procedure of retrieving and genderizing author names is reliable because it produces results close to those expected for a larger, more general dataset."

*Q1.2 In the same line, there are attempts to draw correlations between different fields (i.e. life sciences), which may oversimplify the unique dynamics of each scientific field. However, the lack of more data specific to geosciences in general and seismology in particular explains the comparison with other fields – even if I would prefer to see some more cautiousness in the next stage. In fact, perhaps this limitation of the interpretation / comparisons would deserved to be better highlighted.*

Thank you for this comment. It is indeed true that the lack of studies that address gender inequality in the field of seismology makes it harder to draw field-specific comparisons. Increase in similar studies would be ideal to rigorously address the reasons for gender disparities. We consider that the academic careers in many, if not all, fields follow similar rules and assessment procedures (number of publications, h-index, networking leading to collaborations and more publications, etc). This allows us to make comparisons to different scientific fields. We would also like to emphasise that comparing our results to general trends in various academic fields is helpful. This is because there potentially exist systemic problems that impede the progression of the academic career of women researchers, and gender inequality studies can help tackle them. We have added the following text in lines 260 - 265 to better explain our motivation to compare our results to those of similar studies in other academic fields:

"Throughout this discussion, we compare our findings to the literature on gender gaps in authorship in STEM and other fields of research. Although direct comparisons to other academic fields could be simplifying, they provide an indication of the general trend of women's under-representation in authorship and its subsequent effects in academic career progression. Moreover, they allow us to illustrate some of the possible consequences of authorship gender gaps that have not been studied in geosciences yet. The comparisons should be read as a motivation to conduct further studies that aim to address and quantify gender inequities in the field of seismology or geophysics, rather than as a quantitative discussion."

*Q1.3 Correlations with and extrapolation from EGU data is an important asset of the paper, as it represents an important reliable data specific to geosciences and seismology. However, EGU data is rather complex. There are membership datasets, registration at General Assembly (GA)*

*datasets, and there have been changes on how data is collected (i.e. gender changed from optional or mandatory field a few times…). The last two years of GA were also severely impacted by COVID-19 restrictions (online vs. onsite vs hybrid) which adds other layers of complexity to its data. In conclusion, I would strongly encourage the authors to pursue comparisons with EGU data but either in a dedicated chapter or even paper.*

We thank the reviewer for this comment, which actually reveals some shortcomings that we did not discuss in enough detail. We now separated the description of high-end productivity from the description of comparing EGU demographics with our data. This allows us to highlight the limitations of the comparison more clearly. We think that the comparison is important to show that our approach produces reasonable results when compared to self-declaration. However, ultimately, both datasets measure slightly different things (society membership vs. authorship). Because of that, we have also adapted the first recommendation in the conclusion section, adding:

"As representation in professional societies and conference attendance are not equal to publication productivity, journal publishers should also consider collecting inclusive and self-declared demographic data." (Lines 403 - 305)

Once the EGU dataset has been collected consistently for 5 or 10 years, it will definitely merit a more detailed investigation. It will provide more comprehensive gender diversity data than what is currently available and what we are able to provide, particularly as it is self-declared and more inclusive than automated genderization. The reviewer also alerted us to an inaccuracy here; we are actually considering membership data, which we confused with GA registration data (because attendees often renew their membership during abstract submission or registration). This inaccuracy has now been corrected. The new section is (lines 218 – 234):

"3.6 Towards comparing demographic and bibliographic data

Since 2016, the European Geosciences Union (EGU) systematically collects self-declared demographic information from participants upon membership registration. In 2016–2017, response rates to the question about gender were low, around 50 % for overall seismology section attendees and around 40 % for early career scientists (ECS). The response rates increased in 2018–2019 to around 60 % (ECS: around 50 %) before reaching close to 100 % in 2020.

Between 2018 and 2021, 29–33 % of all seismology section members and 35–38 % of the early career members identified as women. For the years with higher response rates, and for both levels of seniority, EGU seismology members have a larger probability of being female than manuscript

authors in our dataset. The EGU demographics refer to unique members of the seismology community, while our overall probabilities are computed for authorships, i.e. one person may appear repeatedly. The discrepancy in women's participation might indicate a gender gap in authorship. However, several points prevent a direct comparison of both datasets: (i) the results of automatic genderization are less reliable than self-declaration; (ii) while we expect a large overlap of the researchers represented in both datasets, section members may be more frequently working in European countries than article authors; (iii) our dataset can distinguish first authors (who are often, but not necessarily early-career researchers) from other authors, while EGU considers self-declared ECS; these two populations are not directly comparable. Considering these limitations, we find some consistency between our results and the section membership data, namely that ECS members / first authors are more likely to be women than overall members / authors, and that the rate of women membership / authorship up to 2020 is between 20 and 30 % for all members / authors. We cannot conclude that a gender gap in publication productivity exists due to the mentioned caveats, but based on our results such a gap is possible."

*Q 1.4* *The paper conclusions are interesting and the conclusions are of interest for all other fields . Especially, the last bullet "Those evaluating research performance should remain aware that there are, as of now, gender gaps in high-productivity, solo, and high-impact authorship in seismology. "point that deserves a large dissemination. The authors could go even further in terms of ambition in the set of recommendations.*

We thank the reviewer for this positive comment. We moved the recommendation further up, so that we can refer to it in the next point that urges readers to find the causes of these gaps. Also, we extended the recommendation to point out more clearly what the consequences could be:

"Those evaluating research performance should remain aware that there are, as of now, gender gaps in high-productivity, solo, and high-impact authorship in seismology. If this is not taken into consideration in funding and hiring decisions, it may contribute to perpetuating the leaky pipeline problem." (Lines 408 – 409).

---

## Author Comment (AC2)

**Referee #2: Benjamin Fernando**

*SUMMARY:*

*This is an excellent paper which provides a sound evidential basis for an issue of under-representation which affects the entire geosciences community. Only a few minor changes (mostly further explanation or clarification) before being ready for publication in my opinion.*

*We thank the reviewer for the positive, detailed and insightful feedback.*

*TECHNICAL CHANGES:*

1. *line 24 - 'role models' -> 'a lack of role models'?*

   We added the "a lack of" to this sentence.

2. *line 67 (and elsewhere) - is the apostrophe for separating numbers house style?*
   Following the journal style, we corrected the apostrophe to a space, as recommended by the brochure of the SI system. Thank you for pointing this out.

3. *Section 3- did I miss AAGR being defined somewhere?*
   The AAGR stands for average annual growth rates and the abbreviation is defined in line 148.

4. *line 250 (and elsewhere): random capitalisations e.g. 'As'*

   This occurs after a colon; after consulting a style guide we realised that capitalisation of the first word of a full sentence after a colon is an American habit. Consistent with the style of the manuscript, we have removed these capitalisations except where they occur at the beginning of sentences that are part of numbered lists of findings / key points.

*MINOR CHANGES:*

1. *lines 20-21: 'the attrition of female graduates' - there is plenty of evidence for this at a pre-graduate (school) stage too in some of the precursor subjects to seismology (chemistry, physics, maths). I would add in a reference to an appropriate paper to highlight that this is a long-standing, societal issue.*

   Thank you for this comment. We have added several references to the introduction (lines 38 & 39) which cite studies on early-childhood and adolescence resources and their connection to success in science (Hanson, 1997, Dasgupta & Stout 2014, McGuire et al. 2020).

   The problem of the leaky pipeline, however, is a rather complex one that does not necessarily follow a linear progression, i.e. from school years, and development of interest in those stages, to later academic advancement. Some studies show that women participation in PhD programmes is higher than in the BSc/MSc, while their numbers decrease again in later academic stages (e.g. Agnini et al. 2020, for geosciences in Italy). Another complexity regarding the development of interest in STEM fields during childhood and adolescence is presented by Dasgupta & Stout, 2014. They report that this can vary during early and late adolescence in girls and boys. In our study, we consider the stages from graduate education (towards a PhD degree) to subsequent academic progression. Therefore, our data does not allow us to trace the leaky pipeline back to the school years. A study that investigates the connection of upbringing and school subjects precursors to seismology in the later success in this academic field would indeed be interesting. However, such an analysis is beyond the scope of our paper.

2. *line 45: 'assuming a 1:1 gender ratio' - I think that it is worth commenting, even briefly, that a 50:50 assumption produces an underestimate of the scale of the problem as the population ratio isn't quite 50:50.*

   Thank you for raising this point. For the reasons you allude to, we marked this as an assumption, but so far did not comment on it further, because upon closer inspection it is complicated. Public data on the topic frequently refers to sex ratio. Considering global sex ratio in the age cohorts 20, 30, 40 and 50, parity would be approximately 51.5 : 48.5 (male : female, intersex persons are not taken into consideration), as of 2021, meaning that there are in general slightly fewer working age female persons than working age male persons (see https://ourworldindata.org/gender-ratio#sex-ratio-through-adulthood).

We have added a brief footnote to stress the point that this is a simplifying assumption (line 59): "Human sex ratio is generally not exactly 1:1 for male:female persons (intersex persons do not appear in the statistics we consulted). This is because sex ratio depends on multiple factors such as sex ratio at birth, mortality, and selective abortion (e.g. Ritchie and Roser, 2019). This may also affect gender ratio, but how exactly, or whether such data is available, is not known to the authors. Here, we use 1:1 for simplicity."

3. *line 48: 'graduation rates in the US' - is this for just undergraduates, or graduates too? Can you add in a reference for other countries if possible?*

We thank the reviewer for pointing out this line. We revisited this statement and found that it was too generic. In the US, PhD graduation rates of women in geosciences have been relatively high for various years. What we were looking at were undergraduate rates from the 2000's that put women near 45%, which is the lower bound for gender parity used by the UN. Instead of re-formulating the statement here, we have added a more comprehensive paragraph at the beginning of the introduction that summarises data on women in the sciences, STEM, and geosciences. The new paragraph reads (lines 23 - 36):

"Data collected by UNESCO show that while women graduate from any field of study with BSc and MSc at slightly higher rates than men globally (with 53 and 55 % women graduates, respectively), they are slightly less well represented at the PhD level (44 %) and significantly less well represented at the researcher level (29 %), with very pronounced regional variations (Fernandez Polcuch et al., 2018). The European Commission reports that in research in general, the representation of women drops from 52 % at the PhD level to 26 % at the highest career level, while for STEM, 38 % of PhDs and 19 % of senior faculty are women (European Commission, Directorate-General for Research and Innovation, 2021, p. 182).

The leaky pipeline effect was also documented in the geosciences. Agnini et al. (2020) investigated the situation in Italy and reported a drop of women geoscientists from around 50 % at the PhD level to around 20 % at the full professor level (data from 2012 and 2014). Holmes et al. (2008) and Ranganathan et al. (2021) found a similar picture at US-American universities, where approximately 45 % of graduate students and below 15 % of full professors are women. In both Agnini et al. (2020) and Ranganathan et al. (2021), the representation of women faculty is particularly low in geophysics, compared to other geoscience fields. Hori (2020) reported that while women make up 20 % of the Japan Geoscience Union (JpGU) membership, they account for only 2.8 % of JpGU fellows (the

JpGU fellowship is an award bestowed upon senior, accomplished academics). While these are distinct snapshots from specific countries, they all show a consistent pattern."

4. *line 150 (and elsewhere): can you please define exactly what you mean by 'negative bias' - overall representation being poor (<50%), poor relative to another benchmark, or something else?*

Apparently, we did not formulate this clearly, and we thank the reviewer for raising this point. In both sections where this formulation appears, we mean the bias compared to expected values based on the average probability of women authors rather than compared to the 1:1 ratio. We have adapted the text as follows:

Lines 169 & 170:

"Single-authored publications show the strongest under-representation of female-gendered author names, with a negative bias of 6.5 % compared to their expected representation."

Line 208:

"Compared to the overall probabilities, the underrepresentation is most substantial for first authors (negative bias of 7% compared to the average appearance of female-gendered author names), [...]"

5. *lines 149 and line 153: do the 93% and 95% probabilities agree? Or are they different measures? This is unclear to me.*

Thank you for this comment. Here, 93% is the observed probability of finding one woman author when the team size is 12. Conversely, 95% is the ideally *expected* probability if author genders were fully randomly assigned to teams. What we want to point out in line 153 (now line 171 ff) is that a trivial consequence of being a minority is that one is less likely to appear in a sample. The text has been adapted to point this out more clearly (lines 171 - 175):

"**Based on the average probabilities**, a female-gendered author name has a 95 % probability of appearing in a publication when the number of authors is twelve, whereas for male-gendered author names the number of authors required to reach the same probability is three. **In other words, a trivial consequence of women authors being a minority is that only large author teams make their presence highly likely, and only very large teams make it likely that they are not the only one of their gender.**"

6. *line 232: 'corresponds reasonably well' is not a particularly descriptive statement - is this directly in relation to the following lines? If so I would delete this sentence.*

The half-sentence the reviewer is referring to was removed. We agree that it was neither needed nor particularly informative.

*(Slightly more) MAJOR CHANGES:*

1. *line 61: '14 international journals' - can you give a more rigorous description of why you chose these 14? SCOPUS entries? Web of Sciences indices? Line 312 sounds a bit… glib.*

The journals were subjectively chosen based on our experience as seismologists. It is not easy to determine which journals are most important for publishing in seismology; for example, a search for "Geophysics" journals in Scopus would not include Nature, Science, and Nature Geoscience but does include a number of journals that are not particularly important for seismology. We included Nature, Science, and Nature Geoscience because of their prestige. The remaining 11 journals are among the 40 highest cited journals among 186 journals listed in Scopus for the field of Geophysics, and also among the 40 with the highest count of publications.

We adapted the text to make it more transparent that this choice is, at the end, subjective:

Line 75:

"We consider 14 international journals subjectively chosen to cover a broad spectrum of sub-disciplines within the field and a range of impact factors (see Table 1)."

Line 360 – 362:

"We include Nature, Science and Nature Geoscience because of their prestige. The remaining 11 journals are all in the first quartile of journals in geophysics in terms of their number of citations and articles published in 2018-2021 (Scopus)"

2. *lines 114 - 121: the statements that 'we assume the genders of the authors in the article are independent' and 'we derive conditional probabilities of the first author gender [which show correlation]' seem to me to be mutually exclusive - is the mathematics correct in its detail here?*

Thank you for the comment. We assume that the genders of the authors are independent within each individual article but not for the overall conditional probabilities. With the limited information we have regarding the probabilities in each individual article, we think that this is a reasonable approach to computing the overall conditional probabilities. Mathematically, we express this as follows:

$$p(F_1|F_{last}) = \frac{p(F_1 \cap F_{last})}{p(F_{last})} = \frac{\sum_i p_i(F_1 \cap F_{last})}{\sum_i p_i(F_{last})} = \frac{\sum_i p_i(F_1) \, p_i(F_{last})}{\sum_i p_i(F_{last})}$$

We only apply the independence assumption in the last step, while the first two steps are correct regardless of whether or not the genders of the authors are independent. To avoid confusion, we have included all these steps in equation (6) and clarified that the independence is only assumed within each individual article. For the overall probabilities, if the genders of the authors would be independent, then $p(F_1|F_{last}) = p(F_1)$, which is not what our equation shows.

To illustrate that this approach is sensible, consider what would happen if we were to use a threshold and assign binary genders as in Pico et al. (2020): in the rightmost expression of the above equation, $p(F_{last})$ would be 1 for all articles with a woman last author, and 0 otherwise. Then, the expression simply corresponds to the fraction of articles with women first and last authors among all articles with women last authors.

*Figure 2b: what is the statistical significance of the variations from bar to bar (it appears to level off quite quickly, but there's a peak/trough at 8 authors. How many papers have 8 authors and if not many, is a trend line plotted on top also helpful?*

Thank you for pointing this out. The sample size indeed decreases when the number of authors increases. For clarity regarding the statistical significance in each bin, we have modified Figure 2 and indicated the sample size in each case explicitly. We observe that the sample size is below 5% of the total number of articles when the number of authors is larger than seven. Therefore, observed biases in these cases should be carefully interpreted due to their reduced statistical significance. In the previous manuscript version, we were careful to avoid over-interpreting these results. However, it is true that we did not make this transparent for the readers. We have now included a statement clarifying the statistical significance of our results in lines 167 - 169.

3. *Sec 4: Is there another paper that could be written following individual authors (and their publishing trends) through time that could be reported in an anonymised and ethical way? That may be an interesting way of predicting future trends by looking at whether early-career researchers' co-authorship profiles are changing more quickly than those of their more senior colleagues?*

This is an interesting question. We would like to note that for such a study, a dataset composed of more than one decade of published articles would be more appropriate. In this way, career advancement would more adequately be followed, as the process of publication can be relatively long in the field of Seismology (up to one year or more for final publication) and indications of progress can span longer periods of time (than a decade). It is, however, beyond the scope of our paper to study individual researchers' profiles.

Concerning career development as a result of collaborations, and hence co-authorships, there is, to our knowledge, at least one study that investigated women's and men's collaboration patterns with respect to gender, age and internationality in a database consisting of AGU conference abstracts (Hanson et al., Earth and Space Science, 2020). It shows that lack of gender diversity also emerges at the early-career stages and advocates that to tackle gender bias, initiatives to battle the disparities should be considered from the early stages of scientists' careers. Similar studies following the development of individual scientists could be based on such datasets, by bearing in mind that publications can better indicate career advancement than conference abstracts.

4. *Sec 4 (end): I think that the discussion of the limitations of APIs is good, but needs to be more thorough. This is especially true if this is going to be a well-read paper which informs people who are not experts, which I assume it will be. Things to consider: Is there any data or suggestion for what would happen if you treat gender (as a concept rather than a probability) as non-binary? Do we see big changes, or is there just not enough study at the moment? Are there ethnicities or countries for which the chosen APIs are known to perform particularly badly (e.g. I've seen potential suggestion of names from Eastern Asia being particularly poorly sorted).*

Thank you for pointing this out. It is true that our target readers may not be familiar with name-to-gender inference tools used in this study, and a more detailed discussion about their limitations can help interpret our results more appropriately. These tools have two main limitations:

1.- They assign a likely binary gender to each name. Our probabilistic approach intends to reduce this binary logic and consider gender identity as a continuous spectrum with female/male at the two extremes. However, this model does not necessarily represent the actual complexity of gender identities in our society. A more respectful, inclusive, and accurate approach would require using self-declared gender data, which by the time of this study were not available and will likely remain unavailable for several years. We have now added this recommendation in the conclusions: "As representation in professional societies and conference attendance are not equal to publication productivity, journal publishers should also consider collecting inclusive and self-declared demographic data."

2.- They use Latin transcription of names to infer the gender. This approach is particularly challenging for East Asian names, and it has been reported that the accuracy of gender inference tools decreases significantly as a consequence. However, the probabilities assigned to East Asian names are also systematically lower. This means that our probabilistic approach reduces their contribution to our results.

The revised manuscript now includes the following discussion of these limitations in (lines 363 – 377):

"Furthermore, this study does not use self-declared gender data because the journals we consider do not collect and publish them. Instead, we use the term 'gender' to describe the likely perceived gender based on author names using publicly available name-to-gender inference tools (i.e., genderize.io and NamSor). While these tools are frequently used in

similar studies (e.g., Pico et al., 2020; European Commission, Directorate-General for Research and Innovation, 2021), they have two main limitations: (1) They assign a binary gender (female/male) and a probability to each name. By explicitly using these probabilities, we intend to consider gender identity as a continuous spectrum where female/male appear at the two extremes. However, this does not necessarily represent the true complexity and multiplicity of gender identities. (2) They assume that the binary gender of a person can be inferred from Latin transcriptions of their first (genderize.io) or full names (Namsor). Santamaría and Mihaljevic (2018) reported overall inaccuracies of approximately 5% when assessing the performance of genderize.io and NamSor against manually annotated author-gender datasets, with particularly poor performance for Asian names (12% of inaccuracies). Since both gender-inference tools assign systematically low probabilities to Asian names (Santamaría and Mihaljevic, 2018 ), considering gender probabilities rather than fixed thresholds reduces their contribution to our results. This means that misclassifications affect our results less than they would if we were using a fixed threshold. However, it also means that our results reflect the gender distribution of authors with non-Asian names more accurately than that of authors with Asian names."

5. *Data availability - I don't know what the data that the authors are offering to share is - although it is publicly derived, it is worth considering if it is identifiable and if so whether any GDPR constraints (or the like) apply if it is aggregated in a novel way? Not my area of expertise.*

Thank you very much for pointing this out. We looked into it, and indeed it seems that we cannot share the data in its current form, as it includes identifying information, unless we notify the authors. We have adapted the data availability statement accordingly:

"All data used in this study is publicly available. For the convenience of anyone who wishes to use it, the dataset of bibliographic information will be provided after appropriate anonymization (i.e., without author names and article titles) by the corresponding authors upon request."

---

## Author Comment (AC3)

**EDITOR:**

Please relabel the supplementary figures to Figure S1-S3 according to SE guidelines (https://www.solid-earth.net/submission.html#assets > Supplements).

The supplementary material and the references to it in the main manuscript have been adapted accordingly.

**ADDITIONAL SUGGESTIONS FOR CHANGES (following criticism on Twitter):**

We received several critical comments on Twitter after posting the manuscript preprint. Unfortunately, these were not added to the public discussion on EGUsphere, despite our attempt to stimulate such a discussion. Nevertheless, some of these comments appeared to us to point out important issues, that in part have also been raised by Reviewer #2.

Firstly, it was mentioned that the term "female" can be perceived as inappropriate when referring to women. Therefore, we attempted to use the word "women" in place of "female". Wherever we refer to names (e.g., on the figures showing our results), we have kept the terms "female / male", consistently with the genderizing tools, and for simplicity and conciseness.

Secondly, readers pointed out that automatic genderizing on a binary male-female name basis has attracted strong criticism in the past because it makes non-binary and other gender identities invisible and, more importantly, is not based on self-declared gender data (e.g. Strauss et al., 2020). We agree that studies of gender gaps in science should ideally be based on self-declared data, such as those collected by the EGU. We have added corresponding comments in the conclusions. However, such data are currently hardly available or not available at all.

In the manuscript, we have attempted to clarify that we do not use the term "gender" to refer to how people see themselves, but to how they are perceived through their names (lines 363 - 366). We believe that our study is important, because it provides a current approximate view of gender imbalance in seismology authorship and can motivate improvements in data collection in the future.